# Non-Consumptive Effects of *Harmonia axyridis* on the Reproduction and Metabolism of *Spodoptera frugiperda*

**DOI:** 10.3390/insects15060395

**Published:** 2024-05-28

**Authors:** Zeyun Fan, Xiaolu Lv, Yuyang Huang, Weizhen Kong, Chongjian Ma, He Yan

**Affiliations:** 1Engineering Research Center of Biological Control, Ministry of Education, South China Agricultural University, Guangzhou 510640, China; fanzy@scau.edu.cn (Z.F.); lxl17803903494@163.com (X.L.); hxssns@stu.scau.edu.cn (Y.H.); kwz17866928844@163.com (W.K.); 2School of Biology and Agriculture, Shaoguan University, Shaoguan 512005, China; 3Guangdong Provincial Key Laboratory of Utilization and Conservation of Food and Medicinal Resources in Northern Region, Shaoguan 512005, China

**Keywords:** non-consumptive effect, reproductive metabolism, antioxidant enzymes

## Abstract

**Simple Summary:**

This study investigated the non-consumptive effects of *Harmonia axyridis* on the reproduction and physiological status of *Spodoptera frugiperda*. We observed that, even in the absence of predation, the mere presence of *H. axyridis* led to a reduction in the number of eggs laid by *S. frugiperda* adults. This reduction was attributed to the induction of stress responses in *S. frugiperda*, characterized by inflammatory reactions and increased levels of antioxidant enzymes. Alterations in energy allocation occurred, resulting in decreased allocation towards hormone production and vitellogenin synthesis, ultimately leading to diminished fecundity. Our findings underscore the intricate mechanisms through which predators can impact pest populations within agricultural ecosystems.

**Abstract:**

An increasing body of research has underscored the significant impact of non-consumptive effects on the dynamics of prey pests, encompassing growth, development, reproduction, and metabolism across various vertebrate and invertebrate taxa, rivaling the influence of consumption effects. In our investigation, we delved into the non-consumptive effects exerted by the natural predatory enemy *Harmonia axyridis* on the reproductive capacity and metabolism of *Spodoptera frugiperda* adults. Our findings revealed a substantial decrease in the reproductive ability of *S. frugiperda* adults when exposed to the non-consumptive effects of *H. axyridis*. Concurrently, we observed an elevation in hydrogen peroxide (H_2_O_2_) content and the activities of antioxidant enzymes such as superoxide dismutases (SODs), catalases (CATs), and peroxidases (PODs). Furthermore, notable alterations were detected in energy metabolism, characterized by heightened triglyceride levels and diminished glycogen and trehalose concentrations. These outcomes underscored the adaptive response of the pest aimed at mitigating non-consumptive adverse effects by augmenting antioxidant enzyme activity to counteract oxidative stress and minimize cellular damage. Nonetheless, this defensive mechanism entails a significant expenditure of energy resources, resulting in shifts in energy utilization. Elevated triglyceride levels and reduced glycogen and trehalose concentrations diminish available resources for reproductive processes, such as egg laying, ultimately culminating in decreased fecundity. This study contributes novel insights into the non-consumptive effects observed in insects, while also furnishing valuable insights into the mechanisms underlying insect stress responses.

## 1. Introduction

Insects represent one of the most abundant and diverse animal groups on Earth, possessing remarkable adaptability, fecundity, diverse food sources, and multiple modes of spread, collectively contributing to their exceptional survival and stress resistance capabilities [1]. However, their habitats expose them not only to various abiotic stressors, but also to biological stressors [2,3,4]. Abiotic stress primarily arises from environmental factors and human activities, including climate change, environmental pollution, habitat instability, and heavy metal contamination [5,6,7]. In addition to abiotic stressors, insects also face pressures from the biological world, such as predation pressure [8].

There is substantial evidence indicating that non-consumptive effects can induce changes in insect behavior or morphology [9,10], alter the life history of insects [11], and trigger insect immune defense responses and antioxidant responses [12,13]. Moreover, the non-consumptive effects of predatory insects can initiate a trophic cascade by diminishing the feeding and fitness of prey insects [14,15]. It is widely recognized that the non-consumptive effects of natural predatory enemies play a significant role in the detrimental impacts of insect predators on prey insects [16]. In the presence of predators, prey insects typically exhibit defensive behaviors, such as decreasing foraging efficiency and altering foraging behavior, leading to reduced feeding time and decreased adaptability to the pests’ environment [17]. Prey insects are not passive victims of predators. They can detect the presence of natural enemies and employ various cues to continuously adjust their behavior and physiological functions to evade encounters with predators and avoid predation [18]. Therefore, the impact of non-consumptive effects is regarded as an integral aspect of biological pest control strategies [19]. For example, in the case of the leaf-feeding female *Diaphorina citri* Kuwayama (Hemiptera: Psyllidae), when adult *Hippodamia convergens* (Coleoptera: Coccinellidae) are present, the female citrus psyllid secretes less honeydew and reduces deposition and oviposition on plants where ladybirds are detected [20]. Non-consumptive effects can have positive outcomes for prey survival, i.e., changes in prey behavior and physiology that can aid in predation avoidance, while there may also be negative consequences, including increased stress and reduced growth, and even these effects can cascade through trophic systems influencing ecosystem function [15,21]. When prey insects are affected by the non-consumptive effects of natural enemies, there is an increase in metabolic rates in physiological responses and the diversion of energy from other body functions (including antioxidant defense), resulting in elevated oxidative stress [21]. In addition, the reproduction of prey insects can be negatively impacted, leading to reduced reproductive success rates or altered reproductive behavior [22].

*Spodoptera frugiperda* is a major agricultural migratory pest originating from the tropical and subtropical regions of the Americas [23]. Due to its polyphagia and strong adaptability, it has rapidly expanded after invading China, causing serious losses in food production [24]. *Harmonia axyridis* (Coleoptera, Coccinellidae) is not only a well-known aphid predator [25], but also the potential natural predatory enemy of *S. frugiperda* [26]. It has been regarded as a biological control agent [27], which can utilize the perceived sex pheromone of *S. frugiperda* to locate its habitat [28]. However, in addition to the predation ability of *H. axyridis*, little is known about its effects on the biological performance and physiological metabolic function of *S. frugiperda* through non-consumptive effects. In the current study, we investigated the non-consumptive effects of the predatory insect *H. axyridis* on the reproduction, hormones, antioxidant enzymes and energy-related macronutrients of *S. frugiperda* adults. Based on our research findings, we speculate that the non-consumptive effect of *H. axyridis* affects the biological performance of *S. frugiperda* by influencing the aforementioned physiological indicators. The results of this study may contribute to the further development and effective implementation of biological pest control strategies.

## 2. Materials and Methods

### 2.1. Plants and Insects

In this study, maize plants were utilized to rear *S. frugiperda*. Maize seeds (Zhongnong sweet maize 488) were planted in plastic pots measuring 18 cm in diameter and 12 cm × 20 cm in height, filled with a soil and sand mixture comprising 10% sand, 5% clay, and 85% peat. These plants were reared under controlled conditions, excluding other arthropods, and were used in experiments once the maize leaves reached a height of 70–80 cm.

*S. frugiperda* larvae were collected from a maize field in Huadu District, Guangzhou City, Guangdong Province, China, in May 2022. They were subsequently reared on maize plants within a growth chamber (26 ± 1 °C, L/D 14:10 photoperiod, 60–90% photoperiod) for experimental use.

The natural predatory enemy, *H. axyridis*, was reared in the laboratory of the Research Center of the Engineering Research Center of Biological Control, Ministry of Education, South China Agricultural University (SCAU). *H. axyridis* populations were maintained on pea aphid as their primary food source. Prior to the commencement of the experiment, *H. axyridis* individuals were fed with *S. frugiperda* 2nd instar larvae for three consecutive generations to acclimate them to this prey species.

### 2.2. Experimental Setup

In this study, a double-space transparent plastic container (Figure 1) (10 cm diameter and 15 cm height) was utilized as the experimental apparatus. A transparent nylon net (100 mesh cm^−2^) was used to separate the inside of the insect box into two parts. One space of the insect box contained honey water as sustenance for *S. frugiperda*, while *H. axyridis* was introduced to threaten *S. frugiperda* adults in the other space. However, due to the nylon net barrier, the ladybirds were unable to access the other space, resulting in only non-consumptive effects occurring in these experiments.

### 2.3. Non-Consumptive Effects of the H. axyridis on S. frugiperda Reproduction

In order to detect the reproduction of *S. frugiperda* adults under the non-consumptive effect of *H. axyridis*, we introduced one unmated pair of female and male *S. frugiperda* adults into one space of the insect box and introduced four *H. axyridis* adults, within 3 days of eclosion and starved for 24 h, into the other space. After three days of exposure to the threat, the ladybirds were removed. During the stress treatment period, new ladybirds were replaced daily. Throughout the experiment, 8% honey water was provided to *S. frugiperda* adults as sustenance, with fresh honey water replaced daily. Gauze was placed under the lid to serve as a substrate for egg laying. The number of eggs produced was counted daily by shaking them off the gauze. The number of eggs was counted daily until all adults had died. Each adult pair constituted a replicate, with a total of 30 replicates being used. A box with moths but without ladybirds was used as the control.

### 2.4. Sample Preparation and Physiological Indicator Content Determination

Additionally, a stress experiment was set up as described above to measure the activities of the antioxidant enzymes and energy-related macronutrients. After three days of exposing *S. frugiperda* to *H. axyridis*, the female *S. frugiperda* were collected in a petri dish and promptly frozen at −20 °C for 10 min to halt biological activity. Subsequently, they were carefully placed on a microscope slide for dissection in 1× PBS buffer (PH 7.4). During dissection, one dissecting needle gently pressed the dorsal plate of the thorax, while the other needle held the external genitalia at the end of the abdomen and delicately pulled them outwards. This method facilitated the removal of the attached digestive system and body fat while preserving the integrity of the ovaries. Six female adults were collected per sample. The ovaries of *S. frugiperda* were then weighed and subsequently frozen in liquid nitrogen before being ground into fine powder. Then, the crude homogenate was obtained in PBS (pH 7.4) and centrifuged at 10,000 rpm for 10 min at 4 °C. The supernatant was carefully collected and stored at −80 °C until further analysis of the physiological indicator content determination. Protein extraction and measurement were conducted following to the manufacturer’s instructions within the commercially available assay kits from Beyotime (Shanghai, China). Additionally, the activities of the antioxidant enzymes (H_2_O_2_, SODs, PODs, and CATs) and energy-related macronutrients (triglyceride, trehalose, and glycogen) were measured according to the manufacturer ’s instructions for each test kit obtained from Nanjing Jiancheng Bioengineering Institute, China. Each treatment or control trial included three replications to ensure the reliability and reproducibility of the results.

### 2.5. Data Analysis

Data analyses were conducted using SAS software (v.8.01). Prior to applying parametric tests, data were assessed for normality using the Shapiro–Wilks test and for homogeneity of variance using Levene’s test. Differences in fecundity, hormone levels, antioxidant enzyme activity, and energy substance content of *S. frugiperda* adults among different treatments were analyzed using the *t*-test. The Duncan test was employed for post hoc analysis to distinguish significant differences in mean values. Data were represented by mean ± standard error of the mean (SEM). In the figures, asterisks indicate statistically significant differences (*p* < 0.05). Graphs were generated using Graphpad Prism 5 (Graphpad, La Jolla, CA, USA).

## 3. Results

### 3.1. Non-Consumptive Effects of Natural Predatory Enemies on the Biology of S. frugiperda

As shown in Figure 2, the number of eggs laid by *S. frugiperda* exhibited a significant decrease compared to that in the control group (1098.5 ± 168.41) when the *S. frugiperda* adults were exposed to the non-consumptive effect (845.37 ± 157.4) (t_58_ = 6.02, *p* < 0.0001). It can be seen that in the presence of *H. axyridis*, the fecundity of *S. frugiperda* was significantly reduced due to non-consumptive effects.

### 3.2. Non-Consumptive Effects of S. frugiperda Adults on Hormone Activities and VTG Content

The non-consumptive effect of *H. axyridis* markedly influenced the activities of ecdysones (ECRs), juvenile hormones (JHs), and vitellogenin (VTG) in *S. frugiperda* adults. Following exposure to non-consumptive effects, the ECR activity significantly decreased by 29.48% (t_4_ = 3.35, *p* = 0.0285) (Figure 3A), the juvenile hormone significantly decreased by 36.03% (t_4_ = 4.93, *p* = 0.0079) (Figure 3B), and the VTG content significantly down-regulated by 16.14% (t_4_ = 7.06, *p* = 0.0021) (Figure 3C) compared to those in the control group. These results indicated that the non-consumptive effects induced changes in hormone and vitellogenin content in *S. frugiperda* adults.

After exposure to the non-consumptive effects of *H. axyridis*, the activities of the H_2_O_2_ and antioxidant enzymes (CATs, SODs, and PODs) in *S. frugiperda* adults exhibited significant changes, as depicted in Figure 4. H_2_O_2_ activity was significantly up-regulated by 39.11% (t_4_ = −4.38, *p* = 0.008) (Figure 4A) under the non-consumptive effect compared to that in the control group. The non-consumptive effect of *H. axyridis* positively influenced the activity of the antioxidant enzymes in *S. frugiperda*. Specifically, the activities of the CATs significantly increased by 37.43% (t_4_ = −8.21, *p* = 0.0012) (Figure 4B), and the activities of the PODs significantly increased by 54.82% (t_4_ = −10.26, *p* =0.0005) (Figure 4C). However, the activity of the SODs did not exhibit significant change (t_4_ = 1.52, *p* = 0.2031) (Figure 4D). These results indicated that in the presence of *H. axyridis*, the content of reactive oxygen species (ROS), namely H_2_O_2_, increased in *S. frugiperda* due to the non-consumptive effects. In order to counteract the oxidative damage caused by the non-consumptive effects, the activities of the antioxidant enzymes (CATs and PODs) in *S. frugiperda* were significantly up-regulated.

In Figure 5, the non-consumptive effect of *H. axyridis* had a significant effect on the energy-related macronutrients in *S. frugiperda* adults. After exposure to non-consumptive effects, the glycogen in the prey was significantly reduced by 42.62% (t_4_ = 8.2, *p* =0.001) (Figure 5A). Although the protein content increased under the non-consumptive effect, the difference did not reach a significant level (t_4_ = −2.66, *p* = 0.0563) (Figure 5B). Moreover, compared with the control groups, the trehalose content in *S. frugiperda* adults was significantly reduced by 20.68% (t_4_ = 2.78, *p* = 0.0497) (Figure 5C). In addition, the triglyceride content in the prey was significantly increased by 11.41% (t_4_ = −3.22, *p* = 0.0322) (Figure 5D). Our results revealed that the non-consumptive effect subsequently influenced energy-related macronutrients within the *S. frugiperda* adults’ bodies. Specifically, the triglyceride content significantly increased, moreover, while the trehalose and glycogen content decreased in the prey under the non-consumptive effect.

## 4. Discussion

Non-consumptive effects of predators on prey are an important topic in insect ecology. Non-consumptive effects have the potential to alter host prey survival, reproduction, population growth, and so on [29,30,31]. For example, when *Culex pipiens* (Diptera: Culicidae) was faced with the non-consumptive effects of the natural enemy *Notonecta sellata* (Heteroptera: Notonectidae), immature *C. pipiens* developed slowly, adult reproduction was affected and delayed, and the reproduction rate was also significantly reduced [22]. Similarly, Wen and Ueno (2021) found that when *Laodelphax striatellus* Fallén (Hemiptera: Delphacidae) was exposed to an environment with cues of natural enemies *Paederus fuscipes* Curtis (Coleoptera: Staphilinidae), the development and reproductive performance of *L. striatellus* adults deteriorated, and the number of eggs laid also decreased [32]. Moreover, when fruit fly *Drosophila melanogaster* adults were exposed to the parasitoid wasp *Leptopilina boulardi* (Hymenoptera: Figitidae), *D. melanogaster* adults were able to recognize younger wasps as a higher level of threat and consequently depress the oviposition to reduce the risk of parasitism in their offspring, yet *D. melanogaster* did not show a defense response to old parasitoids [10]. Our study also revealed that *S. frugiperda* adults significantly decreased their fecundity when exposed to *H. axyridis*.

Besides reproductive changes, physiological changes may also occur in prey exposed to non-consumptive effect. Juvenile hormones (JH) and ecdysone (20E) are crucial regulators in insect growth and reproduction, and jointly regulate the metamorphosis development of insects [33]. The titers of JH and 20E within insects are affected by various external factors, including temperature, chemical agents, ultraviolet radiation, and heavy metal stress [34,35,36]. For example, different temperatures and photoperiods significantly affected JH titer in *Helicoverpa armigera* (Lepidoptera: Noctuidae) [37]. Additionally, exposure to cadmium changed the larval body size and weight of *D. melanogaster*, prolonged the pupation and eclosion time, and disrupted the level of JH and 20E [38]. Meanwhile, *Spodoptera litura* (Lepidoptera: Noctuidae) exposed to non-consumptive effects showed elevated levels of JH and 20E to varying extents, and the increase in the JH titers enhanced the reproductive activities of adults, while the increase in 20E titers promoted the immune activities of adults [39]. Vitellogenin synthesis is fundamental for vitellogenesis, egg maturation, and embryonic development, directly impacting insects’ fertility, and vitellogenin also plays non-nutritional roles by functioning as immune-relevant molecules and antioxidant reagents [40]. JH regulates vitellin synthesis, oocyte maturation, and the uptake of ooxanthine by oocytes [41]. 20E, a steroid hormone synthesized and secreted by the insect’s prothoracic gland, is vital for regulating growth, development, metabolism, and reproduction [42]. In our study, we found that the non-consumptive effect of natural enemies resulted in reduced levels of JH, 20E, and vitellogenin. We speculated this was the reason for decreasing the fecundity of *S. frugiperda*.

Reactive oxygen species (ROS), as important signal molecules within organisms, have an essential role in various physiological processes [43]. When organisms experience stress, a large amount of endogenous ROS will be produced [44]. Failure to effectively counteract these ROS can lead to significant damage to essential cells and disrupt normal development in insects [43]. In response to rapid ROS accumulation, insects activate an antioxidant defense mechanism, including the increased activity of antioxidant enzymes, to mitigate oxidative damage and maintain cellular homeostasis [45]. For example, studies on the tarantula *Pardosa pseudoannulata* (Araneae: Lycosidae) have shown increased activities of antioxidant enzymes (such as peroxidases, superoxide dismutases, and catalases) in response to cadmium exposure, indicating their vital role in antioxidant defense [46]. Similarly, when *Aphis gossypii* Glover (Homoptera: aphididae) was exposed to the non-consumptive effects of the ladybird *Cheilomenes sexmaculata* (F.) (Coleoptera: Coccinellidae), the level of antioxidant enzymes (SODs and CATs) increased in the body in a short period of time, helping to counteract the oxidative damage induced by elevated malondialdehyde (MDA) content [47]. Therefore, when *S. frugiperda* adults were exposed to the non-consumptive effects of *H. axyridis*, the content of H_2_O_2_ in their bodies increased, indicating the development of severe oxidative stress. In order to reduce the oxidative damage caused by ROS, the levels of antioxidant enzymes (CATs and PODs) increased in *S. frugiperda*.

Exposed to non-consumptive effects, prey may experience stress and respond physiologically by altering their metabolic rates [48]. Triglycerides (TGs) are the main storage form of lipids in the body fat of insects [49], playing a crucial role as an energy source [50]. Similarly, protein mainly synthesized in the body fat is an important basis for insect life substances, and the various life activities of the organism have its participation [51]. Polysaccharides, including trehalose and glycogen, represent the main carbohydrate forms in insects, storing indispensable nutrients and energy while also participating in the metabolism process [51,52,53]. A study by Kaplan et al. [54] showed that when prey organisms were exposed to the presence of predatory insects, their body fat content increased as an adaptive response to store energy, but when the risk subsided, they would convert energy into use. The General Stress Paradigm (GSP) predicts that non-consumptive effects activate physiological stress responses in prey causing diversion of resources away from growth, development, and reproduction to maintain a higher metabolic state [55]. In a prior study, the C content of *Melanoplus femurrubrum* (DeGeer) (Orthoptera: Acrididae) did not change significantly when the species was exposed to chronic non-consumptive effects, but body C/N ratios increased when reared under chronic predation because they increased N excretion [55]. Similarly, Redinger et al. (2022) [56] explored how the elemental and macronutrient content along with the morphology of three abundant Ozark glade grasshopper species differed between glades with and without predatory collared lizard populations, and their results indicated that lichen grasshoppers (*Trimerotropis saxatilis*) increased body C/N ratios in response to predators to heighten metabolism. However, the water flea *Daphnia magna* accelerated growth rates in the presence of a fish predator, reducing their C/N body composition ratios by investing in higher fecundity [57]. In our study, when *S. frugiperda* adults were exposed to the non-consumptive effects of *H. axyridis*, there was a significant decrease in the level of glycogen and trehalose in the body, accompanied by an increase in the level of triglyceride (TG). These results indicated that after *S. frugiperda* adults were stressed by *H. axyridis*, a part of the carbohydrates in the body was converted into fat to maintain the energy and nutrient supply required for life activities. These discrepancies were likely due to different taxa adopting different strategies in response to non-consumptive effects. These adaptive metabolic responses underscore the dynamic interplay between energy storage and utilization in insects facing stress.

To foster the non-consumptive effects of predators that may lead to pest suppression, it is essential to understand what influences prey perception of risk. Cues that induce prey perception of risk may be chemical, with both non-volatile (cuticular hydrocarbons) and volatile organic compounds (VOCs) being used by prey to detect predator presence [29]. For example, small brown planthopper *Laodelphax striatellus* (Fallén) adults exposed to predator cues or predator body extracts of *Paederus fuscipes* Curtis gained less weight and laid fewer eggs [32]. Prey can also use tactile and visual cues to sense predator presences [14] or distinguish vibrational cues from predators vs. non-predators [58]. For example, the tympanic membranes of nocturnal moths can detect bat-like ultrasound frequencies from 10 to 100 kHz, but they are better at detecting frequencies between 20 and 50 kHz [59]. And fruit fly *D. melanogaster* adults perceived parasitic wasps primarily by visual cues [10]. In this study, we utilized double-space transparent plastic containers separated by a transparent nylon net, allowing *S. frugiperda* to sense *H. axyridis* through chemical and visual cues. However, further experiments are needed to determine which cue was crucial.

In summary, the interplay between predators and prey has been a central focus of scientific inquiry within the realm of natural ecosystems over recent decades [60]. A trade-off between reproduction and immunity may occur in insects. An increase in immune function could lead to a decrease in reproductive activity due to the allocation of lesser energy to reproduction, and conversely, infection and activation of the immune system could reduce reproductive output [61]. There was a tradeoff between reproduction and immunity in *S. litura* when exposed to non-consumptive effects, and energy was preferentially allocated to immunity, resulting in decreased reproductive activity and fecundity [39]. In our study, non-consumptive effects led to increased antioxidant enzyme activity, heightened carbohydrate consumption, and decreased fecundity. These findings aligned with previous studies. We speculated that when *S. frugiperda* was exposed to non-consumptive effects, there was induced oxidative stress and cellular damage in the body, prompting an increase in antioxidant enzyme activity to mitigate these injuries. However, the synthesis of these enzymes demanded a significant allocation of energy resources. Consequently, *S. frugiperda* may modulate its metabolic pathways, resulting in heightened consumption of carbohydrate substances. Moreover, to ensure survival in adverse conditions, it may convert surplus energy into fat for long-term storage and energy reserves. This reallocation of resources consequently diminished the energy available for hormone synthesis. Consequently, levels of hormones, including juvenile hormone (JH) and 20-hydroxyecdysone (20E) decreased, leading to reduced synthesis of vitellogenin. As a consequence, the fecundity of *S. frugiperda* adults diminished. In short, a tradeoff between reproduction and immunity was observed in *S. frugiperda*, where energy was preferentially allocated to immunity, resulting in a decrease in fecundity.

Understanding the indirect consumption relationship between natural enemies and their prey is crucial for optimizing the potential of biological pest control [16,62]. Traditional screening of biological control agents often focuses solely on the consumptive abilities of natural enemies. However, non-consumptive effects can significantly induce behavioral, physiological, morphological, and life-history changes in their prey, potentially leading to notable shifts in population and community dynamics [63]. Many studies have discovered that even when natural enemies kill relatively few prey or hosts, they can exert substantial impacts through non-consumptive effects [64,65,66]. Non-consumptive effects may function as effective biological control agents over a larger spatial scale, as pests can detect the presence of a predator from a distance and may move to avoid them, thereby reducing crop damage [16]. Non-consumptive effects can also be directly harnessed for biological control. For example, bat ultrasound has been shown to effectively control agricultural pests, such as armyworms and desert locusts by deterring their intrusion into crop fields [67]. Similarly, in another system, non-volatile cues such as the physical presence of silk collected from the spider predator *Tetragnatha elongata* (Araneae: Tetragnathidae) have been found to reduce plant damage by Japanese beetle *Popillia japonica* (Coleoptera: Scarabaeidae) and Mexican bean beetle *Epilachna varivestis* (Coleóptera: Goccinellidae) prey [68]. In addition, longevity was shorter and fecundity and weight gain were lower when small brown planthoppers *L. striatellus* were exposed to predator *P. fuscipes* body extracts, indicating the possible application of these extracts for pest control [32]. In conclusion, advancements in this field will enhance our understanding of insect stress responses and provide targeted recommendations for the application and refinement of biological control strategies. Future efforts should aim to maximize both the consumptive and non-consumptive effects of predators to achieve more effective biological control.

## Figures and Tables

**Figure 1 insects-15-00395-f001:**
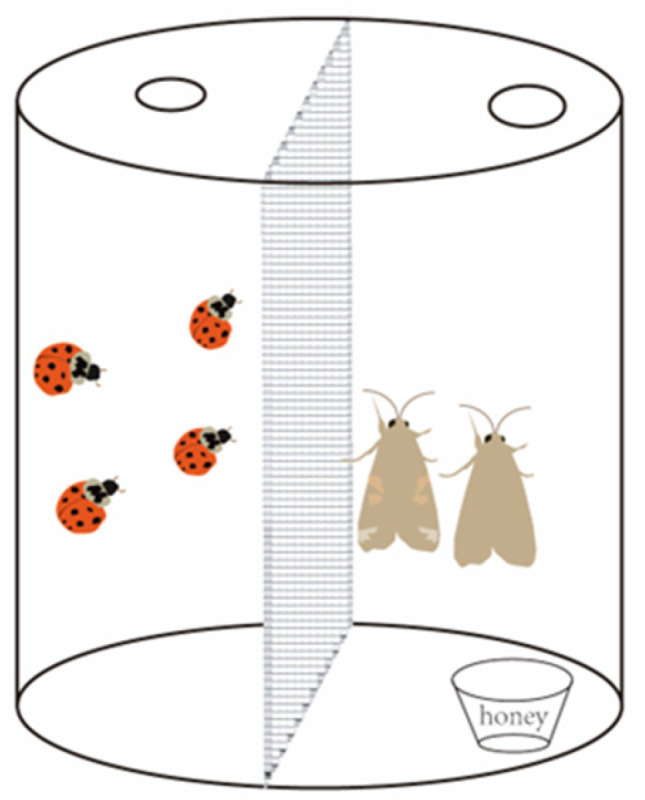
Experimental apparatus for testing the impact of the non-consumptive effects of the *H. axyridis* on *S. frugiperda* adults.

**Figure 2 insects-15-00395-f002:**
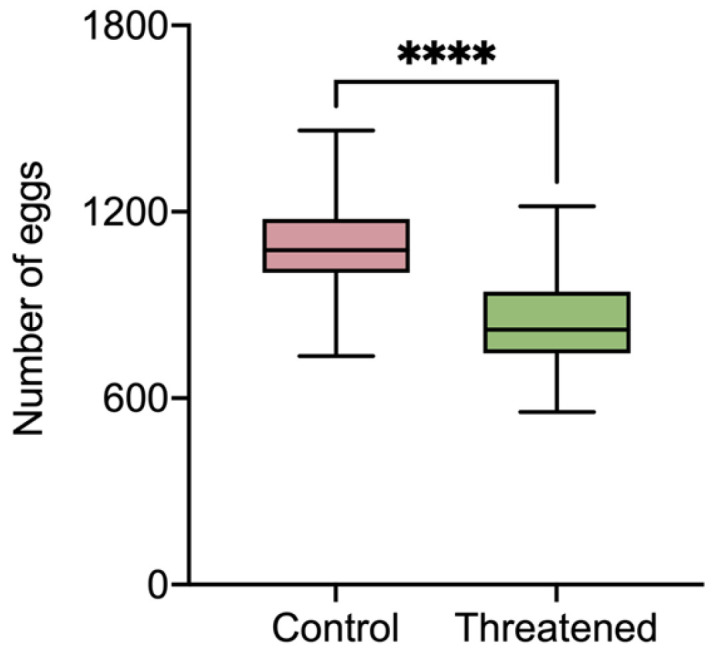
The fecundity of *S. frugiperda* females that were threatened by *H. axyridis*. Those without non-consumptive effects were used as the control. Data are presented as mean ± SE, and **** above the bars indicates a significant difference (*p* < 0.0001).

**Figure 3 insects-15-00395-f003:**
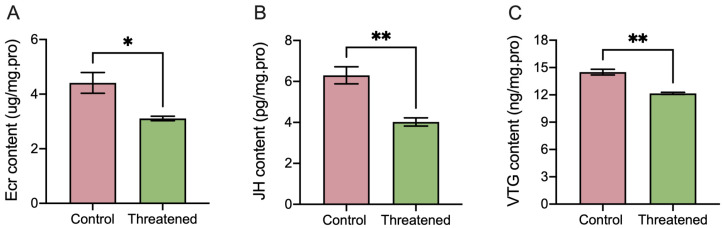
Non-consumptive effects of *H. axyridis* on hormone activities and VTG content of the *S. frugiperda* adults. (**A**) Ecdysone (ECR), (**B**) juvenile hormone (JH), and (**C**) vitellogenin (VTG). Data are presented as mean ± SE, * above the bars indicates a significant difference (*p* < 0.05), and ** above the bars indicates a significant difference (*p* < 0.01).

**Figure 4 insects-15-00395-f004:**
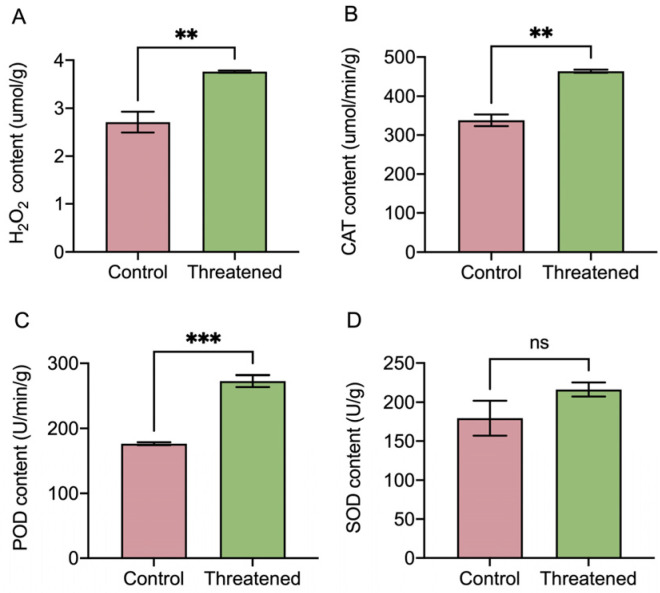
Non-consumptive effects of *H. axyridis* on antioxidant enzyme activities of the *S. frugiperda* adults. (**A**) H_2_O_2_, (**B**) CATs, (**C**) PODs, and (**D**) SODs. Data are presented as mean ± SE, ** above the bars indicates a significant difference (*p* < 0.01), *** above the bars indicates a significant difference (*p* < 0.001), and ns above the bars indicates a significant difference (*p* > 0.05).

**Figure 5 insects-15-00395-f005:**
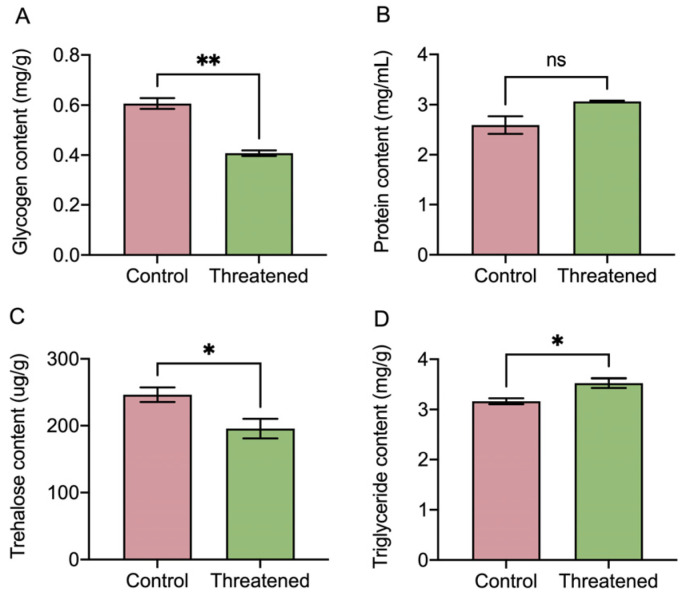
Non-consumptive effects of *H. axyridis* on nutrients in *S. frugiperda*. (**A**) Glycogen, (**B**) protein, (**C**) trehalose, and (**D**) triglyceride. Data are presented as mean ± SE, * above the bars indicates a significant difference (*p* < 0.05), ** above the bars indicates a significant difference (*p* < 0.01), and ns above the bars indicates a significant difference (*p* > 0.05).

## Data Availability

The original contributions presented in the study are included in the article. Further inquiries can be directed to the corresponding authors.

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
