# Peer review of "Non-Consumptive Effects of Harmonia axyridis on the Reproduction and Metabolism of Spodoptera frugiperda"

_insects, 2024, doi:10.3390/insects15060395_

Round 1
Reviewer 1 Report
Comments and Suggestions for Authors
Manuscript insects-3009185 by Fan et al. describes results of a laboratory study of non-consumptive effects of ladybeetles on physiology and reproductive output of fall armyworms. This is an interesting and important subject that has both theoretical and practical significance. The study was properly designed and executed. The manuscript is generally well written, although some further proofreading would be nice. In particular, the term “non-consumptive” should be used consistently throughout the text. My only major comment is that methods should be described in more detail as stipulated below. I also have several minor comments.
Lines 47-49. This sentence is awkward. Non-biological stressors are abiotic.
Line 94. Rear, not rare.
Line 99. Which life stage was collected? Larvae?
Lines 124-125. Was honey water provided for the lady beetles or for the moths? Or both?
Lines 126-127. This sentence is awkward and needs to be reworded.
Lines 129-131. This is unclear. In the previous subsection, it was said that eggs were collected until the death of the individuals. Does this mean the death due to be frozen after three days?
Lines 166-168. What do the dots indicate? Values for each replication? Why are they absent from other graphs?
Line 192 and 219. Non-consumptive, not non-consumable.
Lines 225-226. This sentence is awkward and needs to be reworded.
Comments on the Quality of English LanguageLight proofreading is needed.
Author Response
Response to comments on manuscript insects-3009185
Dear Sir/Madam,
Thank you for the response received from the Reviewers on our manuscript. Please now find enclosed a revised version of the manuscript. We fully appreciate all the constructive comments received from the Reviewers. We have taken on board and revised the manuscript carefully according to the comments and suggestions received were appropriate. We believe the revision has greatly added to the quality of the manuscript.
The Reviewer comments are now listed below with our response given to each one.
I trust this revised manuscript now proves satisfactory. All co-authors are in agreement with the revised version.
Yours sincerely,
Point 1:Manuscript insects-3009185 by Fan et al. describes results of a laboratory study of non-consumptive effects of ladybeetles on physiology and reproductive output of fall armyworms. This is an interesting and important subject that has both theoretical and practical significance. The study was properly designed and executed. The manuscript is generally well written, although some further proofreading would be nice. In particular, the term “non-consumptive” should be used consistently throughout the text. My only major comment is that methods should be described in more detail as stipulated below. I also have several minor comments.
We have taken on board and revised the manuscript carefully according to the comments and suggestions. We believe the revision has greatly added to the quality of the manuscript. And we have consistently revised the term “non-consumptive” throughout the text. Please see line 33, 40,64, 88, 126, 173, 177, 195, 200, 207, 210, 223, 225 and 258.
Point 2:Lines 47-49. This sentence is awkward. Non-biological stressors are abiotic.
Done. We revised sentence. Please see line 46-47. And revised Non-biological to abiotic. Please see line 47, 49.
Point 3:Line 94. Rear, not rare.
Done. Please see line 97.
Point 4:Line 99. Which life stage was collected? Larvae?
Yes. S. frugiperda larvae were collected. We have revised details in line 103.
Point 5:Lines 124-125. Was honey water provided for the lady beetles or for the moths? Or both?
Honey water provided for the moths. During the period of stress, new ladybirds were replaced daily. And we revised details in line 130-132.
Point 6:Lines 126-127. This sentence is awkward and needs to be reworded.
Done. Please see line 135-136.
Point 7:Lines 129-131. This is unclear. In the previous subsection, it was said that eggs were collected until the death of the individuals. Does this mean the death due to be frozen after three days?
We apologize for the misunderstanding. We additionally set up a stress treatment for the fall armyworm according to the methods mentioned in the biological treatment. After the treatment was completed, we sampled to measure antioxidant enzymes and other parameters. We have revised details in line 138-139.
Point 8:Lines 166-168. What do the dots indicate? Values for each replication? Why are they absent from other graphs?
We have redrawn Figure 2-5, and we deleted the dots. Please see revised Figure 2-5.
Point 9:Line 192 and 219. Non-consumptive, not non-consumable.
Done. Please 200 and 225.
Point 10:Lines 225-226. This sentence is awkward and needs to be reworded.
Done. We have revised this sentence in line 232-237.

Reviewer 2 Report
Comments and Suggestions for Authors
Fan et al. describe the effects of exposure to a coccinellid predator (H. axyridis) on egg laying and physiology of Spodoptera frugiperda (Lepidoptera). I am concerned by various aspects of the study and cannot recommend that it be published in Insects.
Main points:
1. The study appears to have been performed with tiny sample sizes (although this is not entirely clear). The authors mention 3 biological replicates (=3 insects?) and two technical replicates (=cages?). The sample size and replication should be clarified and likely greatly increased to provide robust statistical support for the findings.
2. The Discussion is particularly weak. There are few if any insights into the mechanisms of the observed effects. The role of predator volatiles is not mentioned.
3. The text of the Discussion is written in the simplest of terms for issues that have been understood for many decades (e.g. the role of macronutrients on energy metabolism in insects). Modern advances are largely overlooked.
4. The text also provides a selection of examples of predator effects on prey from other systems, but does not focus on proximate (mechanisms) or ultimate (evolutionary) causes except in the most superficial of manners.
5. The contribution of the study to pest control is stated but not described or justified in any detail.
6. As degrees of freedom were not stated for t-tests it was not possible to determine the degree of replication in the study. This needs to be corrected.
Information is missing from some of the references.
7. I have made corrections and queries on the PDF file attached.

Comments on the Quality of English LanguageSee comments written on manuscript.
Author Response
Response to comments on manuscript insects-3009185
Dear Sir/Madam,
Thank you for the response received from the Reviewers on our manuscript. Please now find enclosed a revised version of the manuscript. We fully appreciate all the constructive comments received from the Reviewers. We have taken on board and revised the manuscript carefully according to the comments and suggestions received. We believe the revision has greatly added to the quality of the manuscript.
The Reviewer comments are now listed below with our response given to each one.
I trust this revised manuscript now proves satisfactory. All co-authors are in agreement with the revised version.
Yours sincerely,
Fan et al. describe the effects of exposure to a coccinellid predator (H. axyridis) on egg laying and physiology of Spodoptera frugiperda (Lepidoptera). I am concerned by various aspects of the study and cannot recommend that it be published in Insects.
Main points:
Point 1:The study appears to have been performed with tiny sample sizes (although this is not entirely clear). The authors mention 3 biological replicates (=3 insects?) and two technical replicates (=cages?). The sample size and replication should be clarified and likely greatly increased to provide robust statistical support for the findings.
We apologize for the mistake. In fact, we considered one pair of fall armyworms as one replicate. We conducted a total of 30 repetitions. Please see line 134-135. Additionally, we have redrawn the figures, please see revised Figure 2. Technical replicates refer to taking duplicate samples to ensure the accuracy of the experimental results during the measurement of antioxidant enzymes, nutrients, etc., using the microplate reader. We have deleted the potentially misleading information.
Point 2:The Discussion is particularly weak. There are few if any insights into the mechanisms of the observed effects. The role of predator volatiles is not mentioned.
We have taken on board and revised the discussion carefully according to the suggestions. We have added the mechanisms, please see line 329-335. We have predator volatiles related detailed information in our manuscript, please see line 313- 318.
Point 3: The text of the Discussion is written in the simplest of terms for issues that have been understood for many decades (e.g. the role of macronutrients on energy metabolism in insects). Modern advances are largely overlooked.
We have taken on board and revised the discussion carefully according to the suggestions. We have added the role of macronutrients on energy metabolism in insects in line 290-303.
Point 4: The text also provides a selection of examples of predator effects on prey from other systems, but does not focus on proximate (mechanisms) or ultimate (evolutionary) causes except in the most superficial of manners.
We have taken on board and revised the discussion carefully according to the suggestions. We have added mechanisms in line 328-349.
Point 5:The contribution of the study to pest control is stated but not described or justified in any detail.
We have taken on board and revised the discussion carefully according to the suggestions. We have added the contribution of the study to pest control, please see line 350-374.
Point 6:As degrees of freedom were not stated for t-tests it was not possible to determine the degree of replication in the study. This needs to be corrected. Information is missing from some of the references.
We have added degrees of freedom, please see line 173, 183, 184, 185, 193, 196, 197, 198, 209, 211, 212 and 214. And we have revised the information of all references.
Point 7: I have made corrections and queries on the PDF file attached.
We have taken on board and revised the manuscript carefully according to the comments and suggestions. We believe the revision has greatly added to the quality of the manuscript.
We revised keywords, please see line 40.
We revised “transmission” to “spread”, please see line 45.
We revised “response” to “responses”, please see line 54.
We revised “regards” to “regarded”, please see line 65.
We revised “converens” to “convergens”, please see line 67.
We revised italic, please see line 68.
We reword sentence 69-71, please see line 69-73.
We revised “primary” to “potential”, please see line 83.
We revised “nutrient substance” to “energy-related macronutrients”, please see line 90.
We use “maize” throughout the manuscript, please see line 97, 103.
We revise “were” to “was”, please see line 107.
We added instar. Please see line 111.
We added dimensions of the box and the construction material, please see line 114-115.
We revised “were” to “was”, please see line 115.
We added detail about H. axyridis, please see line 128.
We revised sentence 126-127, please see line 135-136.
We revise “threatned” to “exposed to”, please see line 139-140.
We revised “energy substance” to “energy-related macronutrients”, please see line 155.
We set three replications in each treatment or control trial, and we deleted the potentially misleading information. Please see line 157-158.
We reprocessed the data and redrew all the figures, we revised “CK” to “Control”, “treat” to “threatened”, please see revised Figure 2-5. And we provided df values for all t-test, please see line 173, 183, 184, 185, 193, 196, 197, 198, 209, 211, 212 and 214.
We revised “indiate” to “indicate”, please see line 177.
We revised “N. lugens” to “L. striatellu”, please see line 231.
We reword sentence line 225-226, please see line 232-237.
We revised sentence line 227-228, please see line 237-238.
We revised sentence line 242-244, please see line 259-260.
We revised “is” to “has”, please see line 262.
We revised “is” to “are”, please see line 281.
We reword sentence 271, please see line 289.
We revised acknowledgments, please see line 386-387.

Round 2
Reviewer 2 Report
Comments and Suggestions for Authors
The authors have addressed my concerns in their modified manuscript.
Comments on the Quality of English LanguageThe English requires editing prior to publication.